# Heterogeneity of Zika virus exposure and outcome ascertainment across cohorts of pregnant women, their infants and their children: a metadata survey

Mabel Carabali [iD],[1,2] Lauren Maxwell [iD],[3,4] Brooke Levis [iD],[5] Priya Shreedhar [iD] [4]

For numbered affiliations see end of article.

**Correspondence to**
Dr Mabel Carabali;
mabel.carabali@utoronto.ca

## ABSTRACT

**Objectives** To support the Zika virus (ZIKV) Individual Participant Data (IPD) Consortium's efforts to harmonise and analyse IPD from ZIKV-related prospective cohort studies and surveillance-based studies of pregnant women and their infants and children; we developed and disseminated a metadata survey among ZIKV-IPD Meta-Analysis (MA) study participants to identify and provide a comprehensive overview of study-level heterogeneity in exposure, outcome and covariate ascertainment and definitions.

**Setting** Cohort and surveillance studies that measured ZIKV infection during pregnancy or at birth and measured fetal, infant, or child outcomes were identified through a systematic search and consultations with ZIKV researchers and Ministries of Health from 20 countries or territories.

**Participants** Fifty-four cohort or active surveillance studies shared deidentified data for the IPD-MA and completed the metadata survey, representing 33 061 women (11 020 with ZIKV) and 18 281 children.

**Primary and secondary outcome measures** Study-level heterogeneity in exposure, outcome and covariate ascertainment and definitions.

**Results** Median study sample size was 268 (IQR=100, 698). Inclusion criteria, follow-up procedures and exposure and outcome ascertainment were highly heterogenous, differing meaningfully across regions and multisite studies. Enrolment duration and follow-up for children after birth varied before and after the declaration of the Public Health Emergency of International Concern (PHEIC) and according to the type of funding received.

**Conclusion** This work highlights the logistic and statistical challenges that must be addressed to account for the multiple sources of within-study and between-study heterogeneity when conducting IPD-MAs of data collected in the research response to emergent pathogens like ZIKV.

## STRENGTHS AND LIMITATIONS OF THIS STUDY

⇒ We document the presence of cross-study and within-study heterogeneity across 54 different Zika virus-Individual Participant Data Meta-Analysis (ZIKV-IPD-MA) participant studies, from 20 countries or territories. At the time of submission, our study included one of the largest samples of women (n=33 061) and ZIKV-confirmed cases (n=11 020).

⇒ Study and laboratory protocols changed substantially over time, as knowledge about ZIKV evolved, resulting in important heterogeneity across studies. The in-depth evaluation of the cross-study heterogeneity in exposure, outcome and covariate ascertainment and definitions, elucidate logistic and methodological challenges inherent to conduct an IPD-MA of emergent pathogens.

⇒ The level of heterogeneity across studies underlines the urgent need to develop statistical methods that account properly for the measurement error in both, exposure and outcome ascertainment, in the context of an IPD-MA of an emergent pathogen.

⇒ This assessment could be used to appropriately separate measurement versus clinically relevant sources of heterogeneity in the ZIKV-IPD-MA.

⇒ This appraisal can be used as a template for the transparent and comprehensive treatment of different sources of bias when harmonising and analysing IPD collected in the research response to other emerging pathogens.

## INTRODUCTION

In 2016, the sharp surge in microcephaly incidence led to the declaration of Zika virus (ZIKV) as a Public Health Emergency of International Concern (PHEIC). Five years later, our understanding of the short and long-term effects of ZIKV exposure in utero is still incomplete.[1–4] ZIKV is principally transmitted by *Aedes* mosquitoes and can also be transmitted through unprotected sexual intercourse with a ZIKV-infected partner.[5 6] To date, 90 countries have reported autochthonous ZIKV transmission.[7]

Diagnosis of ZIKV infection is challenging, expensive and time-consuming, especially in

endemic areas where cross-reactivity and cocirculating arboviruses are important concerns.[8–14] Most ZIKV infections (75%–85%) are asymptomatic.[1 5–7] If symptoms arise, the presence of conjunctivitis and rash distinguish ZIKV from other self-limited febrile arboviral infections (e.g., dengue virus (DENV), chikungunya (CHIKV)).[6 15 16] The confirmation of ZIKV includes ruling out the presence of DENV and other arboviruses via real-time or standard Reverse Transcription PCR (RT-PCR) or plaque reduction neutralisation test (PRNT).[9] The performance of both immunological and molecular ZIKV tests differs according to the sensitivity and specificity of the assay, the characteristics of the sample (e.g., sample type, sample maintenance), the timing between symptom onset and sample collection and between sample collection and testing and the implementation and adherence to laboratory quality control (QC) procedures.[8–10 17]

ZIKV infections during pregnancy have been associated with severe outcomes including fetal loss and congenital Zika syndrome (CZS), a spectrum of clinical conditions including central nervous system-related congenital malformations (e.g., microcephaly) as well as ophthalmologic and musculoskeletal abnormalities.[1 3] American and Asian ZIKV strains have been causally related to fetal and postnatal onset microcephaly and CZS; recent nonhuman experimental studies describe higher than expected fetal pathogenicity among recent ZIKV African strains.[18] There is no specific treatment for ZIKV other than management of symptoms and no ZIKV vaccine has yet been approved for use.[6 7 19 20] At the country level, ZIKV-related healthcare needs and employment loss cost an estimated US$2.3 billion per year and caused average yearly loss of over 44 000 disability-adjusted life years.[21–24]

The ZIKV Individual Participant Data (IPD) Consortium was created in 2017 to bring together ZIKV-focused clinical and social science researchers and other stakeholders to conduct an IPD meta-analysis (IPD-MA)[2 25] of ZIKV-related cohorts and surveillance systems that measured ZIKV exposure during pregnancy or at birth. By harmonising existing IPD across related studies, the consortium will leverage existing data to develop and evaluate prognostic models to inform healthcare providers, pregnant women and couples planning a pregnancy and to improve prevention and control measures through identifying groups at the highest risk of adverse consequences during future outbreaks. In addition to priorities for improved diagnosis, management and prevention of ZIKV infection described in the ZIKV Research and Development Roadmap,[26] the ZIKV IPD Consortium's research priorities are to: (1) identify modifiers of the relation between ZIKV infection during pregnancy and adverse short and long-term fetal, infant and child outcomes, (2) assess the role of measurement error and other sources of bias in study estimates and (3) assess the long-term effects of congenital exposure for children with and without clinical or laboratory signs of congenital infection at birth, with consistent definitions of CZS.[2] The ZIKV-IPD Consortium IPD-MA currently includes 65

cohorts and local surveillance sites from 24 countries and territories in the Americas, Asia, Europe and Africa.[2]

Several logistic and analytic challenges complicate the pooled analysis of retrospectively harmonised IPD from longitudinal studies in the ZIKV IPD-MA. Cross-study and within-study assessment of the risk of adverse fetal, infant and child ZIKV-related outcomes has been limited by: heterogeneity in study design, selection bias, differential ascertainment of important confounders or mediators, limitations in the ascertainment of exposure (e.g., timely diagnosis or incomplete follow-up) and outcomes (e.g., pregnancy losses or terminations) and lack of standardisation in definitions across studies, which were primarily launched - during the acute phase of the ZIKV outbreak in 2015–2016.[1–3 7 11] Therefore, well-characterised metadata are essential to appropriately account for study-level sources of heterogeneity in outcome or imputation models and to further improve the precision of subject-level inference in IPD-MA studies.

To support the ZIKV IPD Consortium's efforts to harmonise and analyse IPD from ZIKV-related prospective cohort studies and surveillance-based studies of pregnant women and their infants and children, we developed and disseminated a survey of study-level characteristics (i.e., metadata). Here, we present the results from the ZIKV-IPD metadata survey to provide a comprehensive overview of study-level sources of heterogeneity related to IPD data sets that have or will be made available for the IPD-MA. The in-depth evaluation of cross-study heterogeneity in exposure, outcome and covariate ascertainment and definitions will be used to inform and develop the analytic methods required to appropriately separate measurement-derived versus clinically derived sources of heterogeneity in the ZIKV IPD-MA. This comprehensive metadata review describes the different sources of bias and heterogeneity in the analysis of ZIKV-related studies. It serves as a template for future work to harmonise and analyse IPD collected in the research response to emerging pathogens, where changes to best practice, definitions and study and laboratory protocols proscribe the use of traditional metadata extraction.

## METHODS
### Study selection
Studies eligible to participate in the IPD-MA and corresponding metadata survey were identified through a systematic search registered on PROSPERO (CRD42017068915) that was subsequently published elsewhere.[2 25] The systematic search, developed in consultation with an information scientist and initially conducted on 8 July 2018, applied Medical Subject Headings (MeSH) and text terms to the Ovid (Medline and Embase) without date, geography or language-related restrictions. Additional studies were identified through consultation with subject matter experts, Ministries of Health (MoH) and regional health authorities (WHO, PAHO) and through a monthly PubMed alert that

applied the following text terms: Zika[Title] AND protocol[Title/Abstract] and Zika[Title] AND cohort[Title]. Longitudinal observational studies or local surveillance systems that (1) conducted laboratory testing for ZIKV during pregnancy or at birth; (2) measured fetal, infant and/or child outcomes and (3) included at least 10 individuals (women or children) were eligible. Details of the systematic search to identify eligible studies, including the search strategy, PICOT criteria, eligibility, and the screening and selection steps are presented in online supplemental appendix 1, figure S1. Eligible studies were invited via e-mail to share data and to actively participate in the development and completion of the IPD-MA.

### Metadata survey development and dissemination

To obtain key metadata information, specifically about procedures that could have potentially changed after studies' implementation, and to obtain information directly from investigators instead of relying on information included in published studies or protocols, we developed a comprehensive metadata survey of completed and ongoing ZIKV-related studies available in English, Spanish and Portuguese (online supplemental appendix 2). We programmed the survey using Research Electronic Data Capture (REDCap)[27] software and disseminated the survey to sites that agreed to share participant-level data for the IPD-MA. The metadata survey included 964 items related to study location, design, population, enrolment criteria and sample size; maternal, placental, fetal and infant ascertainment of ZIKV; ascertainment of microcephaly and CZS (clinical and laboratory criteria), other arboviruses and Syphilis; Toxoplasmosis, Rubella, Cytomegalovirus and Herpes (STORCH) pathogens; genetic testing, child development assessment, funding and duration and timing of follow-up contacts. ZIKV IPD Consortium members (online supplemental appendix 3) formed expert panels that were designated as subject matter working groups (WGs) for outcomes, exposures, social sciences, metadata and harmonisation. Each WG included clinicians with expertise in relevant areas (e.g., neonatology, neurology, psychology, virology), epidemiologists, sociologists and members of the metadata and harmonisation team (i.e., consortium members who will work on the harmonisation of the data as it becomes available). Using the ZIKV IPD-MA protocol as a guide, the WGs met and reviewed the established exposure and outcome-related components of the metadata survey to ensure its comprehensiveness, this is, that the draft survey included all potential exposures, outcomes and likely confounders and the methods for their ascertainment or measurement. The WGs also assessed the utility of survey in terms of public health and epidemiological relevance to potential end users such as other researchers, policymakers and healthcare practitioners. The metadata survey was first circulated to ZIKV-IPD Consortium members in June of 2020 and closed to new responses in March 2021. Investigators received two monthly reminders to complete the survey and were contacted individually to resolve study-level missingness and to address inconsistencies between survey responses and published study reports.

### Data analysis

Overall design, recruitment, exposure, outcome and covariate ascertainment variables included in the resultant metadata dataset were described using median and IQR for continuous variables and proportions for categorical variables, analysis and figures were created using RStudio Team (R Core Team (2020) Version. 4·0·5). The metadata were assessed for cross-study and within-study heterogeneity in recruitment, exposure, outcome and covariate ascertainment and definitions. ZIKV ascertainment practices were assessed by the type (e.g., serological or molecular testing), timing of the ascertainment (e.g., antenatal, perinatal and postnatal) and type or source of guidelines used. We conducted a *post hoc* qualitative bias assessment that describes potential issues with design and exposure and outcome ascertainment.

### Patient and public involvement statement

This study assesses already collected data from ongoing or already finished ZIKV research studies conducted during the ZIKV epidemic and PHEIC. Our unit of analysis are studies, not individuals. For this study, we did not involve patients in the development of research questions or outcome measurement. However, the ZIKV-IPD-MA Consortium aims to leverage existing data to develop and evaluate prognostic models to inform pregnant women and couples planning a pregnancy and to improve prevention and control measures through identifying groups at the highest risk of adverse consequences during future outbreaks. The consortium is in constant communication with principal investigators (PIs), co-PIs and site coordinators of the participant studies. Each participant study has a study-specific plan to disseminate their results. IPD-MA research results are disseminated among PIs, co-PIs and site coordinators as part of the already established different monthly WG meetings and yearly consortium meetings.

### Role of the funding source

The systematic review and metadata survey were drafted and revised with funding support provided by the Wellcome Trust to the WHO Department of Sexual and Reproductive Health and Research Human Reproduction Program, grant numbers 206532/Z/17/Z and 216002/Z/19/Z, respectively. The funder of the study had no role in study design, data collection, data analysis, data interpretation or writing of the report.

### RESULTS

From 160 eligible publications identified through the systematic search and 145 studies identified through consultation, 63 cohort study or surveillance sites, representing 59 unique data sets agreed to share deidentified,

participant-level data for the ZIKV-IPD-MA (figure 1A). The metadata survey was completed for 54 of these data sets (91.5%). Remaining studies (n=5) did not initiate or complete the metadata survey after 9 months of regular follow-up. The four teams that reported a reason for non-participation, reported multiple considerations including inadequate time or resources to support participation (n=4), lack of consensus on participation from their team (n=1) and need for special permissions from MoH (n=1). The difference between the number of study sites and unique data sets is explained by the inclusion of studies using data from a single surveillance system or multisite studies for which the metadata (e.g., source population, ZIKV assays and outcome ascertainment) did not vary across sites. For instance, the NIH/National Institute of Allergy and Infectious Diseases (NIAID)funded International Prospective Observational Cohort Study of Zika in Infants and Pregnancy study[28] and the European Commission Horizon 2020-funded ZIKAlliance study[29 30] each standardised outcomes definitions and laboratory procedures across sites, but due to specific differences in the source population or follow-up procedures, these studies are analysed individually by study site here. In contrast, the US Zika Pregnancy and Infant Registry[31 32] is an enhanced surveillance programme including 50 states and other US territories with a shared standardised protocol contributing to a single metadata data set. The Puerto Rico Zika Active Pregnancy Surveillance System study,[33] part of the overall US Zika registry programme, has a slightly different protocol and is included as a separate data set.

## Recruitment

Table 1 presents the basic characteristics of participating studies by geography. The 54 studies that completed the metadata survey included 33 061 women (non-pregnant, pregnant and puerperal), 11 030 confirmed maternal ZIKV cases and 18 281 infants and children from 20 countries or territories. Details including study location, authors, source population, enrolment characteristics, study design, recruitment and follow-up visits, sample size and funding are presented in online supplemental appendix 1, table S1. Twenty-three studies (43%) were conducted in Brazil (figure 1B). Thirty-seven studies (68%) recruited participants from healthcare facilities and three studies included travellers returning from endemic countries. Fifty studies enrolled pregnant and or puerperal women, three studies enrolled infants at birth and collected retrospective information on ZIKV symptoms during pregnancy, and one study included women of reproductive age regardless of pregnancy status. Figure 2 shows the duration of enrolment and infant and child follow-up by type of funding and figure 3 shows the study size, enrolment start date and ZIKV positivity by study design. Duration of enrolment ranged from 1 to 60 months (median=18; IQR=11, 28 months) (table 1). The most frequent study design was prospective cohort (83%). Surveillance studies (15%), with recruitment based on clinical or laboratory criteria (e.g., PCR or immunoassays), were by default case-only studies and often included larger sample sizes. Studies that limited enrolment to ZIKV positive women (32%) or had a higher percentage of ZIKV-positive women often started enrolment before the declaration of the PHEIC in 2016 (figure 3). Self-funded studies (n=7) and studies receiving local or national funding (n=12) generally initiated enrolment before the declaration of the PHEIC while studies receiving international funding mostly started enrolment months after (figure 2).

## Exposure ascertainment

Assessment of ZIKV infection in pregnant or puerperal women and fetal and infant ascertainment are presented in table 1 and figures 4 and 5. Assessment of maternal ZIKV status (n=51) included laboratory confirmation alone in 38 studies (74%) and clinical definitions plus laboratory confirmation in 13 studies (25%). The median proportion of ZIKV-positive women per study was 47% (IQR=3, 100%). The laboratory assessment of ZIKV in pregnant women included PCR (94%), ELISA for immunoglobulin M (IgM) or immunoglobulin G (IgG) (69%) and PRNTs (24%). From the studies performing PCR, 71% reported cycle threshold (Ct) values and 73% reported QC procedures; 74% of studies conducting ELISAs, reported QC procedures. Four surveillance-based studies where the testing was done at local or national surveillance laboratories did not provide details of laboratory techniques or QC procedures. Twenty-seven studies ascertained fetal ZIKV exposure using a combination of maternal and fetal laboratory criteria (e.g., diverse testing of fetal tissues or umbilical cord) and used imaging as supporting ascertainment tool. Forty-one studies ascertained ZIKV in infants using PCRs (80%), ELISAs (61%) and PRNTs (12%). Maternal testing for DENV and CHIKV was conducted in 69% (n=35) and 61% (n=31) studies, respectively (figure 4).

## Outcome ascertainment

Antenatal and postnatal ascertainment of microcephaly was completed in 51 studies (94%). Criteria used to define microcephaly followed national or international standards (i.e., WHO, Intergrowth or MoH definitions) in 42 studies (81%); 10 studies (19%) used their own study-specific criteria. Congenital ZIKV exposure or CZS was assessed in 46 studies (85%), from which 31 (67%) used international standard criteria and 15 (33%) study-specific definitions. Fetal ultrasounds were performed in 39 (74%) studies (range: 1–12 ultrasounds per pregnant woman) and 10 studies performed fetal MRIs. Postnatal cranial ultrasounds and brain MRIs were performed in 36 and 17 studies, respectively. Termination of the pregnancy was documented in 30 (55%) studies (table 1). Duration of follow-up (median=24; IQR=15, 29 months) differed by the ZIKV status of pregnant women, microcephaly status of newborns and funding source (figure 2).

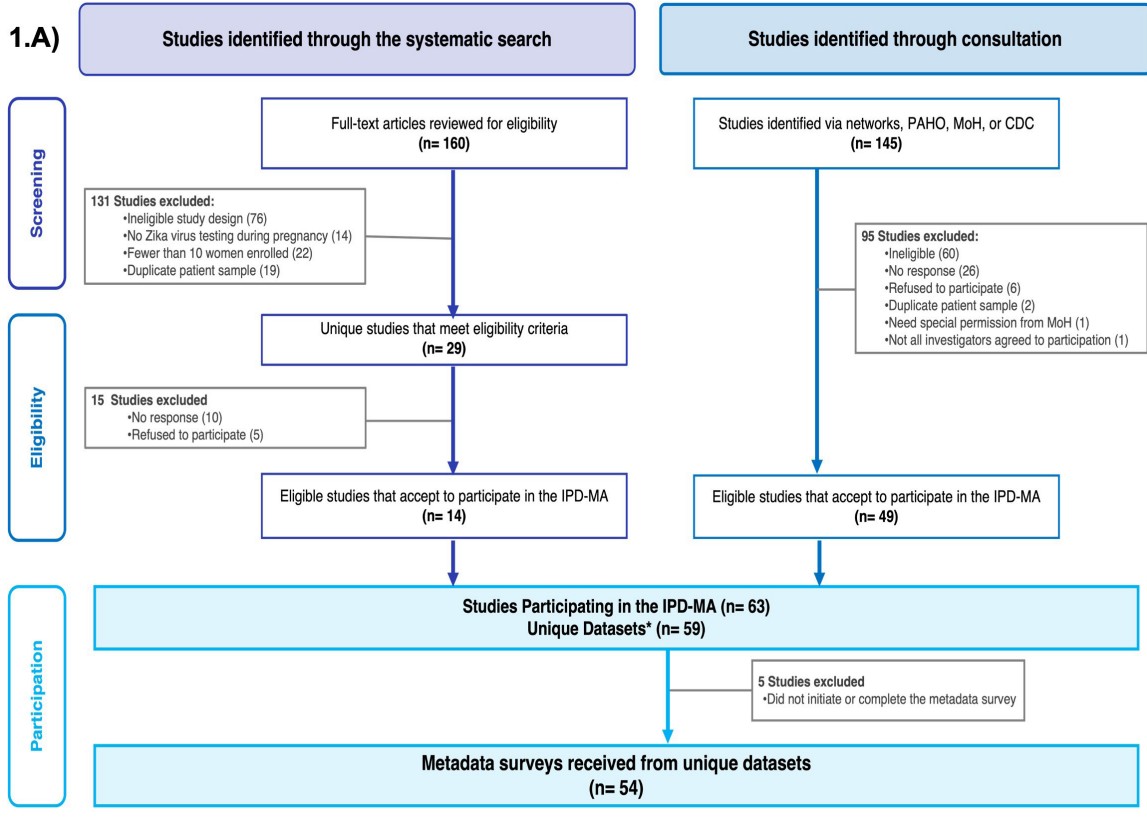

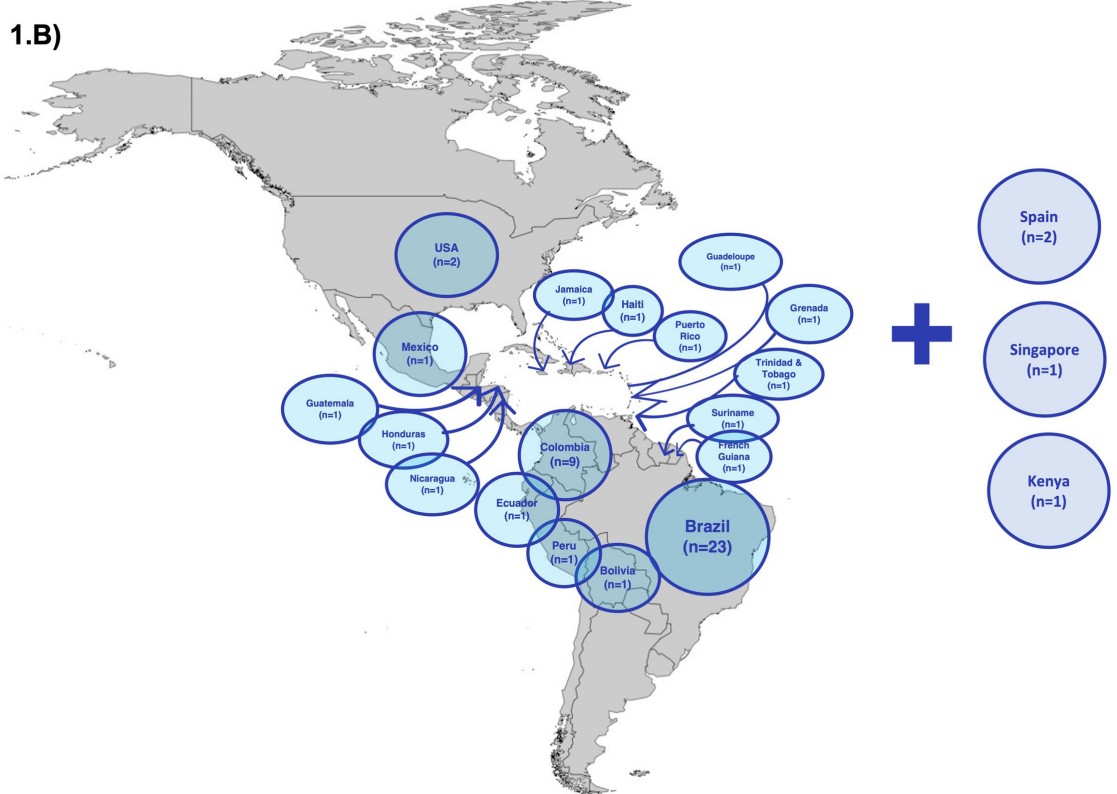

**Figure 1** Metadata flowchart (A) and ZIKV-metadata countries and territories participants (B). *The difference between the number of study sites and unique datasets is explained by the inclusion of studies using data from a single surveillance system or multisite studies for which the metadata (e.g., source population, ZIKV assays, and outcome ascertainment) did not vary across sites. CDC, Center for Diseases Control; IPD-MA, Individual Participant Data-meta-analysis; MoH, Ministries of Health; PAHO, Pan American Health Organization; ZIKV, Zika virus.

**Table 1**  Summary table of study-level characteristics among participants of the ZIKV-IPD-MA metadata survey

| Characteristic | Overall N=54* | Study's origin | |
| --- | --- | --- | --- |
| | | Brazil N=23 (43%) | Other countries N=31 **(57%)** |
| Number of pregnant women | | | |
| Median (IQR) | 268 (100, 698) | 178 (98, 522) | 436 (113, 1,108) |
| Number of ZIKV (+) | | | |
| Median (IQR) | 68 (18, 182) | 66 (47, 150) | 72 (16, 182) |
| % ZIKV (+) per study | | | |
| Median (IQR) | 47 (3, 100) | 68 (7, 100) | 36 (2, 100) |
| Number of children | | | |
| Median (IQR) | 154 (54, 487) | 130 (54, 357) | 220 (52, 616) |
| Study design | | | |
| Case cohort | 1 (2%) | 1 (4%) | 0 (0%) |
| Cohort | 45 (83%) | 20 (87%) | 25 (81%) |
| Surveillance | 8 (15%) | 2 (9%) | 6 (19%) |
| Source population | | | |
| Community | 6 (11%) | 4 (17%) | 2 (6.6%) |
| Community and healthcare facility | 8 (15%) | 6 (26%) | 2 (6.6%) |
| Community+travellers | 2 (4%) | 0 (0%) | 2 (6.6%) |
| Healthcare facility | 37 (68%) | 13 (57%) | 24 (77%) |
| Travellers | 1 (2%) | 0 (0%) | 1 (3.2%) |
| Enrolment criteria | | | |
| Children only | 3 (5%) | 3 (13%) | 0 (0%) |
| Not pregnant | 1 (2%) | 1 (4%) | 0 (0%) |
| Pregnant and/or puerperal women | 50 (93%) | 19 (83%) | 31 (100%) |
| Inclusion limited to ZIKV(+) pregnant women | 17 (32%) | 6 (26%) | 11 (37%) |
| Duration of enrolment (months) | | | |
| Median (IQR) | 18 (11, 28) | 18 (11, 27) | 18 (11, 27) |
| Duration of follow-up (months) | | | |
| Median (IQR) | 24 (15, 29) | 24 (12, 36) | 24 (18, 24) |
| Received funding | 47 (87%) | 20 (87%) | 27 (87%) |
| Type of funding received | | | |
| International funding | 22 (41%) | 4 (17%) | 18 (58%) |
| Multiple sources | 13 (24%) | 10 (43%) | 3 (9.7%) |
| National/local funding | 12 (22%) | 6 (26%) | 6 (19%) |
| No/self-funding | 7 (13%) | 3 (13%) | 4 (13%) |
| Maternal ZIKV assessment | 51 (94%) | 20 (87%) | 31 (100%) |
| Maternal immunoassay | 35/51 (69%) | 13/20 (65%) | 19/31 (61%) |
| Maternal PCR | 48/51 (94%) | 17/20 (85%) | 31 (100%) |
| Maternal PRNT | 12/51 (24%) | 5/20 (25%) | 7/31 (23%) |
| DENV testing | 35/51 (69%) | 15/20 (75%) | 20/31 (64%) |
| CHIKV testing | 31/51 (61%) | 14/20 (70%) | 17/31 (61%) |
| HIV testing | 44/52 (85%) | 19/23 (83%) | 25/29 (86%) |
| STORCH testing | 40/51 (78%) | 19/23 (83%) | 21/28 (75%) |
| Fetal ZIKV assessment | 27/53 (51%) | 10/22 (45%) | 17/31 (55%) |
| Infant ZIKV assessment | 41/53 (77%) | 17/22 (77%) | 24/31 (77%) |

**Table 1** Continued

| Characteristic | Overall N=54* | Study's origin | |
| --- | --- | --- | --- |
| | | Brazil N=23 (43%) | Other countries N=31 (57%) |
| Prenatal microcephaly assessment | 40 (74%) | 16 (70%) | 24 (77%) |
| Pre and postnatal microcephaly assessment | 51 (94%) | 23 (100%) | 28 (90%) |
| Fetal ultrasounds | 39/53 (74%) | 15/23 (65%) | 24/30 (80%) |
| Postnatal ultrasound | 36/53 (68%) | 13/23 (57%) | 23/30 (77%) |
| Fetal MRIs | 10/52 (19%) | 3/23 (13%) | 7/29 (24%) |
| Postnatal MRI | 17/52 (33%) | 8/23 (35%) | 9/29 (31%) |
| CZS assessment | 46 (85%) | 22 (96%) | 24 (77%) |
| CZS identification criteria | | | |
| Study definition | 15/46 (33%) | 7/22 (32%) | 8/24 (33%) |
| WHO/CDC/MoH | 31/46 (67%) | 15/22 (68%) | 16/24 (67%) |
| Genetic screening | 9/53 (17%) | 2/23 (8.7%) | 7/30 (23%) |
| Performed child developmental assessment | 40/53 (75%) | 16/22 (73%) | 24/31 (77%) |
| Documented termination of pregnancy | 30 (55%) | 9 (39%) | 21 (68%) |

*Total number of studies, unless otherwise specified is (n=54 for all studies; n=23 for Brazilian studies and n=31 for other countries).
CHIKV, Chikungunya Virus; CZS, Congenital Zika Syndrome; DENV, Dengue Virus; MoH, Ministry of Health; PRNT, plaque reduction neutralisation test; STORCH, Syphilis; Toxoplasmosis, Rubella, Cytomegalovirus and Herpes Testing; ZIKV (+), confirmed ZIKV positive.

Infant and child development were assessed in 40 (75%) studies using several different international scales, some of which were adapted to local languages and contexts.[34] The most used scales were the Bayley Scales of Infant Development (BSID) (75%) and the Ages & Stages Questionnaires Social-Emotional, Second Edition (ASQ-SE-2) (52%). The age range for developmental assessment varied greatly (median age=24; IQR=18, 30, range=3, 60 months). Two studies from the same research group reported applying the Warner Initial Developmental Evaluation of Adaptive and Functional Skills and four the Alberta Infant Motor Scale to provide further information on developmental and motor delays not captured by the ASQ-SE-2 and BSID[35] (online supplemental appendix 1, table S2). Five studies reported planning to follow children beyond age 5 to assess developmental delays that relate to school performance, using the INTERGROWTH-21st Neurodevelopment Assessment or the neuropsychological tests NEPSY–II.

## Covariate ascertainment

Genetic screening and testing were performed in nine and eight studies, respectively. Online supplemental figure S2 (online supplemental appendix 1) presents the distribution of genetic testing, maternal STORCH and another pathogen ascertainment. HIV testing was conducted in 44 (81%) studies. Fifty-one studies tested for STORCH pathogens, including syphilis (72%), toxoplasmosis (70%), rubella (59%), Cytomegalovirus (57%) and herpes (37%). Four studies tested for malaria. Sociodemographic and behavioural factors measured by studies included education (83%), occupation (62%), marital status (77%), household socioeconomic status (71%), maternal exposure to workplace teratogens (33%), exposure to intimate partner violence (18%), recreational drug use/abuse (82%), smoking (82%), alcohol consumption (80%) and ethnicity (67%) (online supplemental appendix 1, table S3). Covariate selection and ascertainment varied by study type (surveillance-based vs cohort), duration, timing related to declaration of PHEIC, location and funding level and source.

## Quality of data and risk of bias assessment

The metadata data set included information obtained directly from investigators and substantiated by published reports to ensure reliability and quality of the information. The potential sources of bias identified across the participant studies relate to selection bias due to differential selection, as by the presence of symptoms or observable congenital malformations, into the study and follow-up, and measurement error due to exposure and outcome ascertainment which were differential by the presence of symptoms and by ZIKV assay results. Hence, the potential risk of both selection bias and measurement error was likely highest at the beginning of the epidemic, and surveillance-based studies were likely affected by under and over-reporting at different stages of the epidemic. The metadata survey demonstrated that study protocols changed importantly over the course of study implementation to account for advances in the understanding of ZIKV presentation (e.g., the high percentage of asymptomatic cases), diagnostics (e.g., cross-reactivity with DENV), these changes increased the likelihood for potential misclassification, and for the ascertainment of

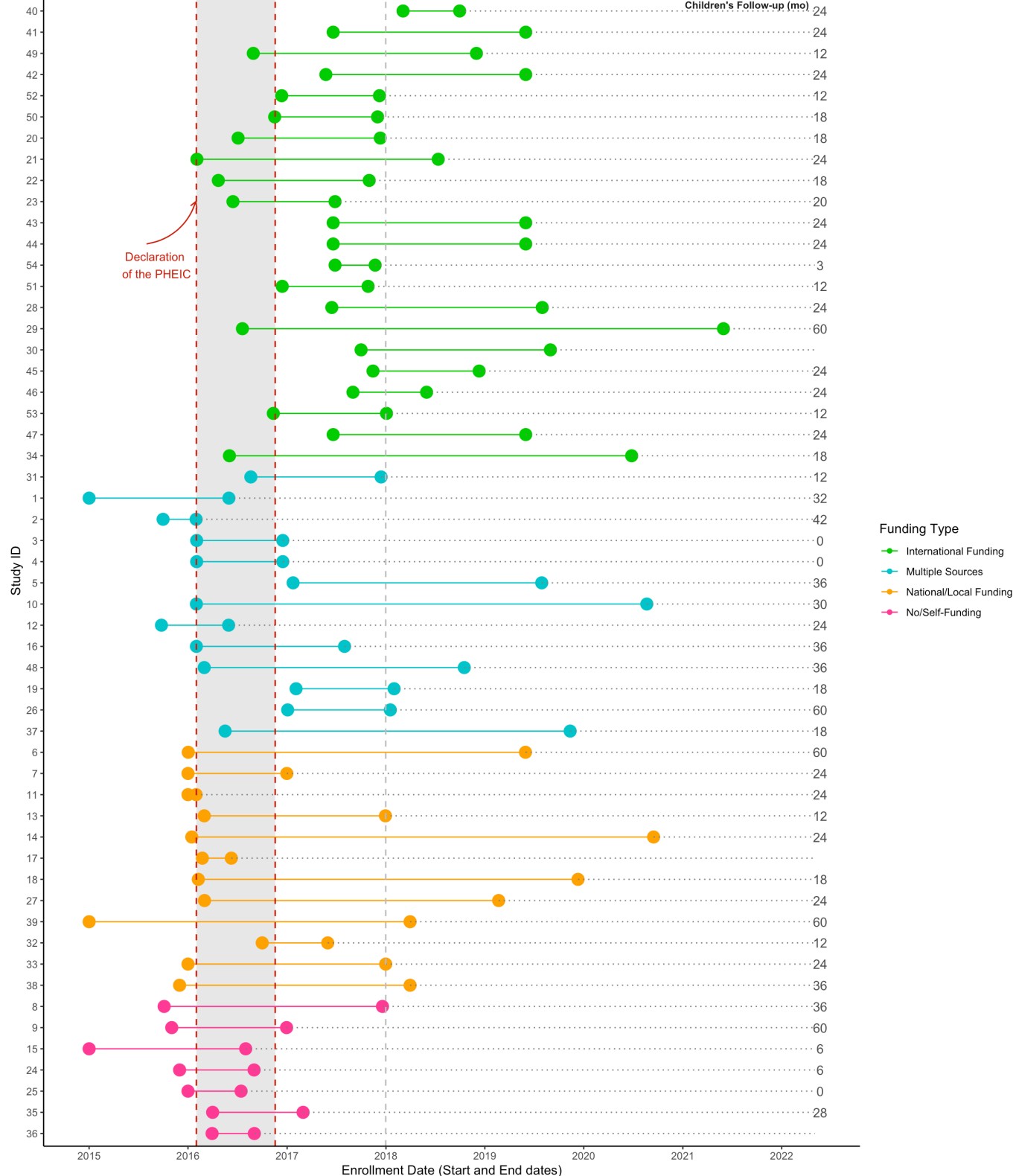

**Figure 2** Duration of enrolment and follow-up for children by type of funding among participants of the ZIKV-IPD-MA metadata survey. IPD-MA, Individual Participant Data-meta-analysis; PHEIC, Public Health Emergency of International Concern; ZIKV, Zika virus.

long-term outcomes (e.g., development of adverse developmental outcomes that were not observed at birth), and to accommodate funding availability, especially in

the Americas region. The misclassification of ZIKV infection is foregrounded in our presentation of the diversity of methods, resources, definitions and classifications

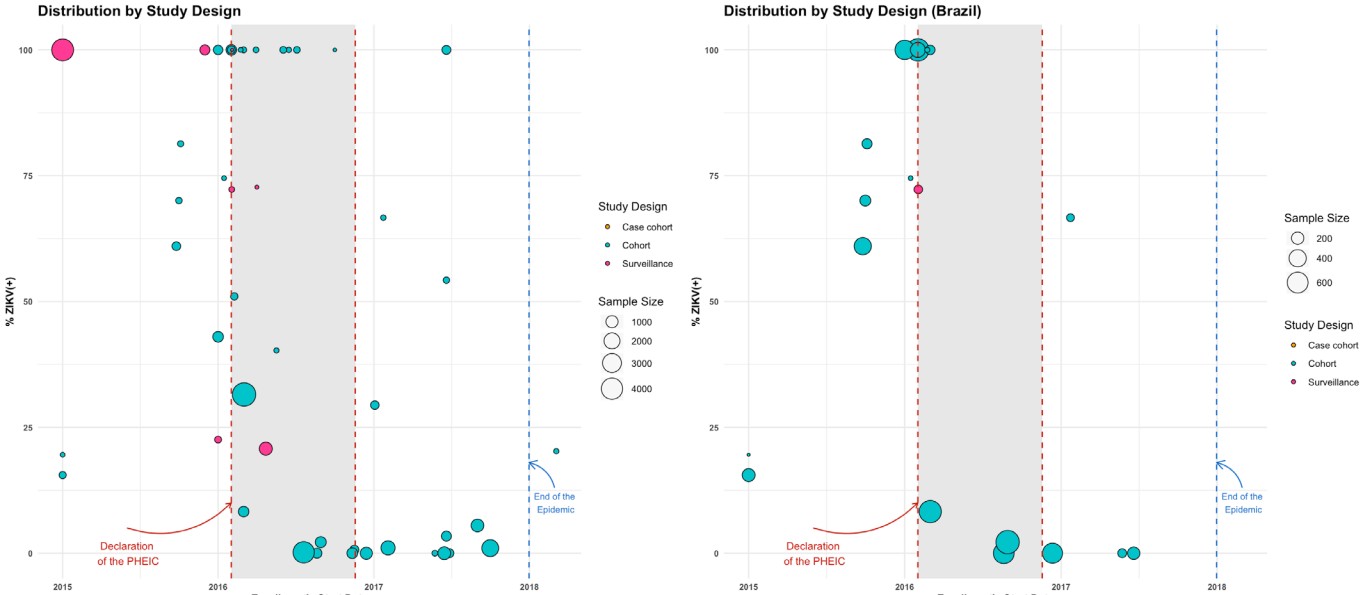

**Figure 3** ZIKV positivity by study's start date, sample size and study design among participants of the ZIKV-IPD-MA metadata survey. IPD-MA, Individual Participant Data-meta-analysis; ZIKV, Zika virus.

applied in studies' ascertainment of maternal, fetal and infant ZIKV infection across all study settings and designs (table 1, figures 4 and 5). Genetic testing, fetal ultrasounds and MRIs used for outcome ascertainment are presented in online supplemental appendix 1, table S4, and the type of maternal, fetal and infant ZIKV exposure assessment among ZIKV-IPD-MA study participants of the metadata survey are presented in online supplemental appendix 1, tables S5a-S5c.

## DISCUSSION

We conducted a comprehensive metadata survey to collect study-level data on exposure, outcome and covariate ascertainment to ensure the appropriate harmonisation and analysis of participant-level data from ZIKV-IPD-MA contributing studies. We found a high level of cross-study and within-study heterogeneity in both maternal ZIKV exposure and outcome ascertainment and the analysis of the evolving spectrum of congenital malformations and postnatal onset developmental delays that define CZS.

The range and distribution of study-level metadata have important public health and epidemiological implications related to the: (1) accurate communication of individual-level and population-level risk, (2) reliable identification of prognostic factors and effect measure modifiers for ZIKV-related outcomes and (3) urgent need to develop statistical methods that account properly for the measurement error in both exposure and outcome ascertainment in the context of an IPD-MA. Below we discuss the advantages of our approach to metadata collection, key study-level sources of heterogeneity and the potential implications of our findings for the analysis of completed or ongoing IPD-MAs and the design of future research responses to emerging pathogens.

First, in the research response to emerging pathogens, stakeholder consultation is likely more effective than a traditional systematic review for identifying eligible studies and obtaining reliable metadata.[36–40] We initially tried to collect study-level metadata from publications or otherwise available protocols. However, we found important differences in design, definitions and ascertainment procedures between the available documentation and investigator self-reports. The conflicting or incomplete information across information sources, especially as related to exposure ascertainment and QC procedures, led us to develop the metadata survey.

Second, our metadata survey identified sources of heterogeneity across and within studies that are not commonly observed in IPD-MAs of well-characterised pathogens or in chronic diseases.[41–43] Study and laboratory protocols changed substantially over time, as knowledge about ZIKV evolved, resulting in important heterogeneity across studies.[5 9 15] For example, diagnostic algorithms and testing guidelines evolved from using serology and RT-PCR to include PRNT testing over the course of the epidemic.[5 9 14 15 44 45] The high ZIKV infection attack rates (17–55%) reported during the epidemic across the Americas and the causal link with microcephaly[4 5 17 46–49] prompted a rapid rollout of cohorts of pregnant women. Due to the natural course of the epidemic and to study inclusion criteria, studies initiated around the epidemic peak or immediately after the declaration of PHEIC had higher ZIKV-positive rates among pregnant women compared with studies initiated later in the epidemic. Enrollment (criteria and duration), the follow-up for pregnant women and their children and the planned methodological approaches varied by the type of funding. Thus, the natural course of the epidemic,

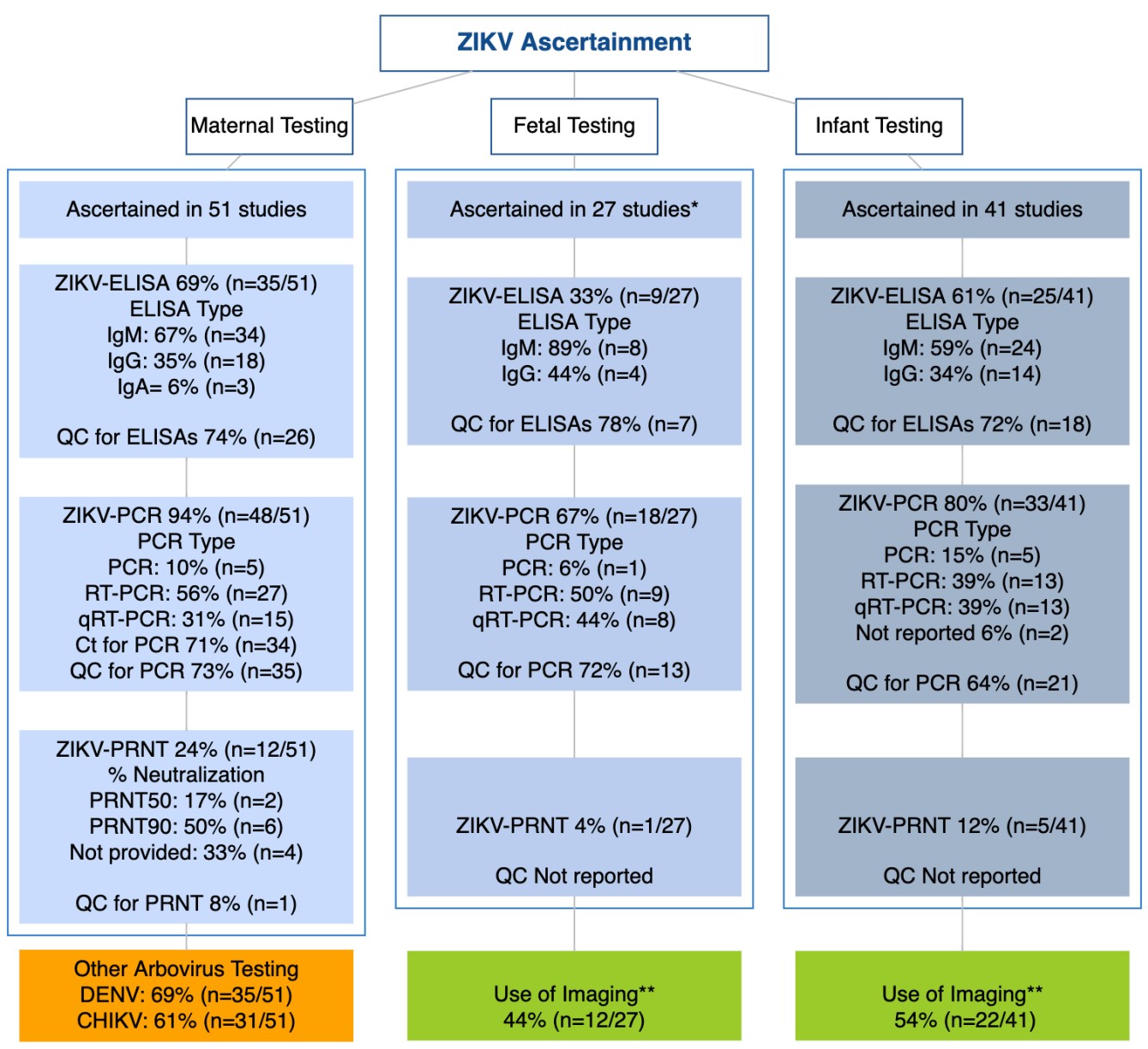

**Figure 4** Overview of ZIKV ascertainment among participants of the ZIKV-IPD-MA metadata survey. CHIKV, chikungunya virus; DENV, dengue virus; PRNT, plaque reduction neutralisation test; IPD-MA, Individual Participant Data-meta-analysis; ZIKV, Zika virus. *Ascertainment of ZIKV included a combination of maternal and fetal testing in some cases. **Imaging was reported as supporting information for the ascertainment.

changes to understanding of ZIKV aetiology and funding mechanisms[50] impacted the quality and amount of information collected, further contributing to cross-study heterogeneity.

Third, ZIKV diagnosis and confirmation is challenging.[3 9–12 14 15 17 45] Our findings indicate that the various criteria, definitions, assays and testing algorithms used for ZIKV diagnosis are key sources of heterogeneity in ZIKV infection ascertainment. The main challenges for the confirmation of ZIKV infection in our survey were the (1) lack of standardised or consistent guidelines and algorithms for ZIKV diagnosis; (2) ascertainment of ZIKV infection in settings that are endemic for other arboviruses using assays that do not adequately account for cross-reactivity and (3) lack of ascertainment of fetal ZIKV infection. Most studies indicated that ZIKV infection among pregnant women was ascertained using both clinical and laboratory criteria. However, the range and scope of the clinical criteria was as diverse as the number of studies and the definition of laboratory confirmation included several combinations of immunoassays and molecular techniques, each one with different standards of procedures, variable or unknown cut-off points and different or unknown approaches for QC. The urgency and limited knowledge about ZIKV called for the simultaneous, asynchronous development of diagnostic tests,

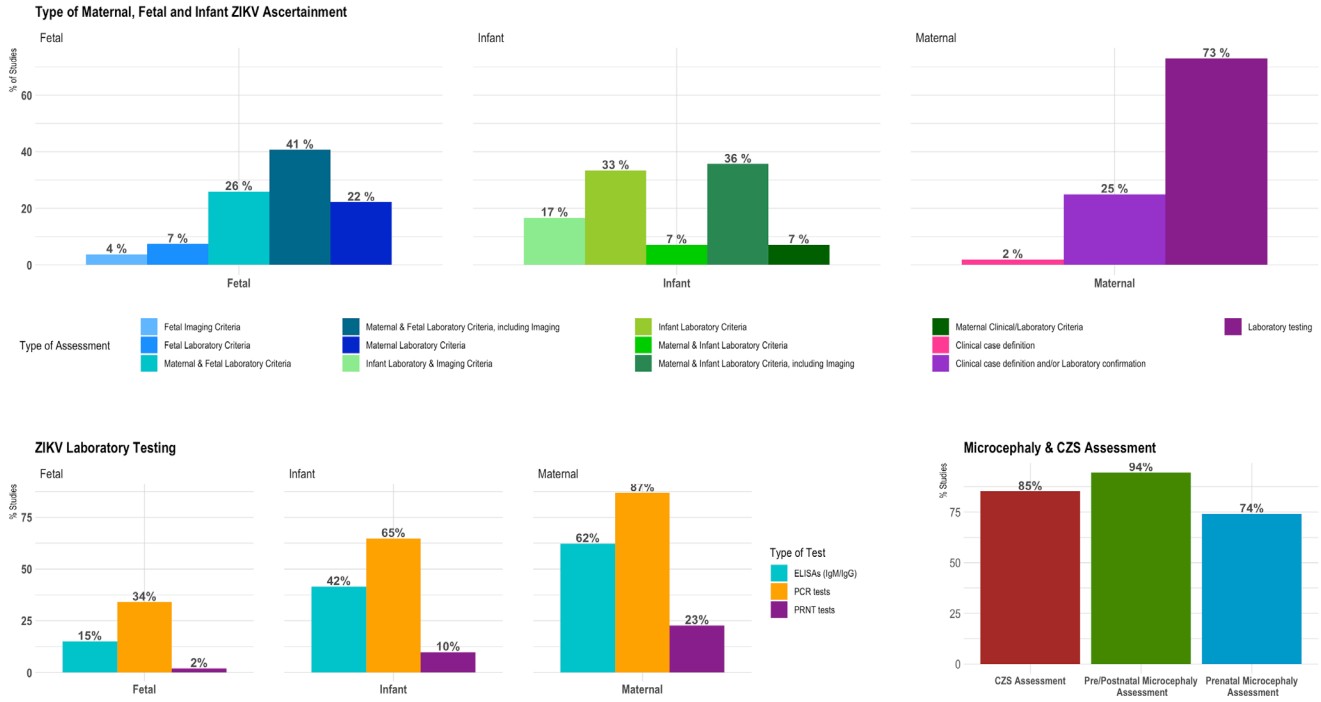

**Figure 5** ZIKV ascertainment and type of laboratory testing and other criteria among participants of the ZIKV-IPD-MA metadata survey. CZS, congenital Zika syndrome; IPD-MA, Individual Participant Data-meta-analysis; ZIKV, Zika virus.

where researchers converged over time to a somewhat smaller number of more accurate assays or testing algorithms.[8 9 14 15 26 31] This lack of standardisation can be seen as both detrimental, if at early stages of the epidemic the testing was based only on the presence of symptoms or if the methods used were less accurate,[11 13–16 31] and beneficial because it led to innovation in test development.[9 10 12 15 17 26]

Fourth, smaller studies that were launched earlier in the epidemic, which were often the less resourced, are potentially affected by an outcome and exposure ascertainment that is differential by the probability of infection, which in turn may influence the estimation of the long-term and short-term risk of ZIKV effects. Surveillance-based studies include mostly symptomatic individuals who are able to seek and receive healthcare attention (e.g., diagnosis and standard care).[51] Given the high proportion of asymptomatic ZIKV infections,[6 7 47 52] some surveillance-based studies are likely affected by underreporting.[15 46 52–54] Neither studies launched early in the epidemic nor surveillance sites included comparable data on women who were not infected with ZIKV during pregnancy. In addition, health-seeking behaviour for symptomatic individuals depends also on socioeconomic determinants of access to health care,[46 53] potentially leading to differential outcome ascertainment, representing another source of bias.[17 53 55] For instance, in endemic areas, the presence of symptoms was related to antenatal healthcare practices (e.g., number of antenatal visits, ZIKV testing and ultrasounds and medical termination of the pregnancies)

during the epidemic,[46] leading to potential differential assessment of exposure and outcomes, including for miscarriages, induced and spontaneous fetal losses and congenital malformations. Additionally, cultural sensitivity and restricted legal environments meant that only half of studies measured termination of pregnancy, which biases results differently across sites depending on abortion access.[56] Furthermore, a wide range of definitions of congenital Zika were implemented. Some studies used the WHO definition of CZS, others a modified version including isolated manifestations like ophthalmological abnormalities and several the presence of microcephaly at birth to define CZS. Therefore, the few cohort studies with longer follow-up are valuable resources for understanding the longer term consequences of congenital ZIKV infection among infants born without manifestations of congenital ZIKV because studies with short or no follow-up for children would not have captured potential cognitive or developmental issues.[1 3 34 35]

These four issues pose both statistical and logistical challenges when harmonising and analysing data in the context of an IPD-MA of an emerging pathogen. The ZIKV-IPD Consortium recognised that existing methods for addressing the correlated sources of heterogeneity in the research response to ZIKV (e.g., dependency between timing of symptoms, test accuracy and differential ascertainment for both exposure and outcome, etc) are inadequate. In response to this gap, the Consortium launched a statistical WG to develop novel approaches to address heterogeneity and measurement error in the

context of IPD-MAs of emerging pathogens. The statistical WG will use the findings from this metadata survey to guide the development of an analytic plan that accounts for the different sources of correlated measurement error. The statistical analysis plan will extend existing methods for measurement error and clinical risk prediction to address the specific challenges of an IPD-MA of an emerging pathogen. Methodological approaches for accounting for correlated sources of measurement error and the high levels of heterogeneity in the research response to emerging pathogens will be context specific and pathogen specific. Although specific recommendations or solutions to account for the diverse sources of heterogeneity are desirable, an in-depth proposal for the methods to address these statistical challenges is outside of the scope of this descriptive analysis.

## Limitations and strengths

The inclusion of ongoing studies presented important challenges in that key aspects of exposure and outcome ascertainment have continued to evolve as studies obtain additional funding to retest samples or to continue and expand follow-up.[50] In some cases, studies changed exposure ascertainment procedures or laboratory QC procedures and were not able to record when changes were made or what had been changed, which represents missing or misclassified metadata, further complicating the assessment of the accuracy of exposure ascertainment.[2 9 17] Although we conducted regular follow-ups over a 9-month period to obtain the information, some researchers did not respond, reasonably because they were on the frontlines of the COVID-19 pandemic response. This impacted the timelines for participant acquisition and metadata acquisition and review for most studies. Nonetheless, the multiple interactions required with the investigators enriched the analytical process, ensured the robustness and quality of the data and strengthened the cross-consortium communication.

The identification of the above-mentioned sources of heterogeneity constitutes the main strength of this study. This is the first study to assemble a comprehensive reporting of study-level heterogeneity of ZIKV-related cohorts of pregnant women and their infants and children, which could only be gathered through investigators' self-report. The final metadata and participant-level data sets will represent the largest resource of longitudinal, participant-level data on maternal-infant/child dyads for many of the countries where research teams have contributed data for the IPD-MA. Ensuring that participant-level data are accompanied by well-characterised metadata, coupled with engaging the teams that collected the data and external subject matter experts, will help the ZIKV IPD Consortium ensure the appropriate estimation of outcome and imputation models. This detailed study-level metadata will facilitate the way that IPD can be harmonised and analysed, including the identification of study-related sources of heterogeneity and clinically related participant-level sources of heterogeneity.[36 37 40] Given

the limited understanding of the distribution of adverse fetal, infant and child outcomes related to congenital exposures, the meta- and participant-level data set will represent a tremendous cross-country and within-country resource for social, medical, infectious and other exposures, that goes beyond ZIKV.

## Conclusion

This analysis provides a comprehensive overview of study-level heterogeneity related to participant-level data available from ZIKV-related cohorts and surveillance systems of pregnant women and their infants and children. The challenges confronted and addressed here include: (1) the need to consult with researchers and local and international health authorities early in the outbreak to identify related studies; (2) the need to engage directly with investigators to understand and document heterogeneity in study design, definitions and implementation, which are inconsistently or incompletely reported in the published literature and include information according to the existing knowledge of the problem at that given time; and (3) the need to consistently and regularly adjust research targets and timelines to account for continuing changes to study protocols, teams, agendas and funding availability, ensuring the quality of the studies. Our study will serve to inform ongoing and future collaborations that prospectively or retrospectively harmonise IPD and to drive the development of novel statistical methods that can address correlated sources of measurement error in IPD-MA of emerging pathogens.

**Author affiliations**
[1]Departement de Médecine Sociale et Préventive, Université de Montréal, Montreal, Quebec, Canada
[2]Department of Epidemiology, Dalla Lana School of Public Health, University of Toronto, Toronto, Ontario, Canada
[3]Department of Sexual and Reproductive Health and Research, World Health Organization, Geneve, Switzerland
[4]Heidelberger Institut für Global Health, UniversitätsKlinikum Heidelberg, Heidelberg, Germany
[5]Centre for Prognosis Research, School of Medicine, Keele University, Keele, UK

**Acknowledgements** The authors would like to acknowledge the valuable contributions of all researchers, fieldworkers and participants of the contributing studies of the ZIKV-IPD-MA.

**Collaborators** The ZIKV-IPD-MA Consortium: Pablo Aguilar Ticona, Luiz Carlos Junior Alcantara, Jackeline Alger, Celia Alpuche Aranda, Jonathan Altamirano, Juan Arias, Lumumba Arriaga-Nieto, Melissa A. G. Avelino, Angel Balmaseda, Azucena Bardají, Carlos Hernan Becerra Mojica, Mónica Benavides, A.P. Bertozzi, Karen Blackmon, Victor Hugo Borja Aburto, Patrícia Brasil, William J Britt, Nathalie Broutet, Pierre Buekens, André Cabié, David Alejandro Cabrera-Gaytán, Rodrigo Cachay Figueroa, María Luisa Cafferata, Juan Ignacio Calcagno, Juan P. Calle, Mabel Carabali, Derrick WS Chan, Celia CD Christie, Federico Costa, Antonio Jose Cunha, Carlos Cure, Johanna Antonia Adriana Damen, Marcela C. Daza, Roberta L. DeBiasi, Thomas Debray, Valentijn M.T. de Jong, Camille Delgado López, Leah deWilde, Alan Oliveira Duarte, Geraldo Duarte, Valorie Eckert, Esther M. Ellis, Andres Espinosa-Bode, Cassia Fernanda Estofolete, Roberta Evans, Valéria Christina de Rezende Féres, Fabíola Fiaccadori, Lester Fernando Figueroa Bolaños, Olivier Fléchelles, Arnaud Fontanet, Victoria Fumadó, Anna L. Funk, Anna Gajewski, Rosa Margarita Gelvez Ramirez, Carlo Giaquinto, Luz Gibbons, Suzanne M. Gilboa, Maria Barbara Franco Gomes, Anna Goncé, Cesar R Gonzalez-Bonilla, Eduardo Gotuzzo, Concepción Grajales-Muñiz, Rebecca Grant, Elysse Grossi-Soyster, Tahani Hamdan, Eva Harris, Cosme Harrison, Bruno Hoen, Cristina Hofer, Natanael Holband, Ivonne Huerta, Irene Inwani, Thomas Jaenisch, Esaú João, Edna Kara, Salma Khuwaja,

Caron Kim, Albert I. Ko, Nancy F. Krebs, Angelle Desiree LaBeaud, Heather Lake-Burger, Ellen H. Lee, Vernon Lee, Yee-Sin Leo, Brooke Levis, Eduardo Lopez-Medina, Anyela Lozano-Parra, Elena Marbán-Castro, Celina Maria Turchi Martelli, Lauren Maxwell, Clara Menéndez, Marcela Mercado Reyes, Conrado Milani Coutinho, María Consuelo Miranda Montoya, Demócrito de Barros Miranda-Filho, Maria Elisabeth Moreira, J. Glenn Morris, Jr., Sarah B. Mulkey, Johanna Munoz, José Esteban Muñoz-Medina, Peninah Munyua, Marisa Marcia Mussi-Pinhata, Nivison Nery Jr., M. Kariuki Njenga, Mauricio L. Nogueira, Eric Osoro, Miguel Parra-Saavedra, Saulo Passos, Bernadete Perez Coêlho, Monika Piccardi, Léo Pomar, Arnaldo Prata-Barbosa, Ingrid Rabe, Mitermayer Reis, Hannah Rettler, Megan R. Reynolds, Ana Maria Rivera Casas, Diana Patricia Rojas, Teresita Rojas-Mendoza, Nicole Roth, Paola Mariela Saba Villarroel, Magda Sanz Cortes, Janet L. Sayers, Deolinda Scalabrin, Lavinia Schuler-Faccini, Stacey Schultz-Cherry, Kirstin Short, Priya Shreedhar, Antônio Augusto Silva, Ronaldo Silva, Isadora Siqueira, Karen Sohan, Carmen Soria-Segarra, Antoni Soriano-Arandes, Patrícia da Silva Sousa, Maria Benamor Teixeira, Claire Thorne, Soe Soe Thwin, Van T. Tong, Benoit Tressieres, Marília Dalva Turchi, Diana Valencia, Miguel Valencia-Prado, Alfonso Vallejos-Parás, Maria Van Kerkhove, Luis Angel Villar, Carmen Viñuela Benéitez, Manon Vouga, Randall Waechter, Yinghui Wei, Jamie Westcott, Marc-Alain Widdowson, George SH Yeo, Ricardo Arraes de Alencar Ximenes. The full list of participants of the ZIKV-IPD-MA Consortium members is attached as supplementary material.

**Contributors** LM, MC and BL were responsible for the study conception and design. LM designed and conducted database searches to identify eligible studies. MC, LM, BL and PS contributed to data extraction and coding for the metadata analysis. MC, LM and BL contributed to the data analysis and interpretation. MC and LM wrote and BL, PS contributed to the first draft of the manuscript; ZIKV-IPD Consortium participants contributed to subsequent versions of the manuscript. All authors provided a critical review and approved the final manuscript. LM, MC, BL and PS are the guarantors; they have full access to all the data in the study and take responsibility for the integrity of the data and the accuracy of the data analyses.

**Funding** This systematic review and summary of metadata was drafted and revised with funding support provided by the Wellcome Trust to the WHO Department of Reproductive Health and Research Human Reproduction Program, grant numbers 206532/Z/17/Z and 216002/Z/19/Z, respectively. Dr. Levis was supported by a Fonds de Recherche du Québec—Santé Postdoctoral Training Award. No other main authors reported funding for primary studies or for their work on the present study. Funding sources for each participating study in the ZIKV-IPD-MA are provided in the supplementary material. The funding source had no role in the drafting of the manuscript. The study funder had no role in study design, data collection, data analysis, data interpretation, or writing of the report. The corresponding author had full access to all data in the study and final responsibility for the decision to submit for publication.

**Disclaimer** WHO: The authors alone are responsible for the views expressed in this article and they do not necessarily represent the views, decisions, or policies of the institutions with which they are affiliated. United States Centers for Disease Control and Prevention: The findings and conclusions in this report are those of the authors and do not represent the official position of the Centers for Disease Control and Prevention (CDC).

**Map disclaimer** The inclusion of any map (including the depiction of any boundaries therein), or of any geographic or locational reference, does not imply the expression of any opinion whatsoever on the part of BMJ concerning the legal status of any country, territory, jurisdiction or area or of its authorities. Any such expression remains solely that of the relevant source and is not endorsed by BMJ. Maps are provided without any warranty of any kind, either express or implied.

**Competing interests** All authors have completed the ICMJE uniform disclosure form and declare: no support from any organisation for the submitted work; no financial relationships with any organisations that might have an interest in the submitted work in the previous three years for the main authors. No funder had any role in the design and conduct of this study; collection, management, analysis, and interpretation of the data; preparation, review, or approval of the manuscript; and decision to submit the manuscript for publication. Consortium members and study principal investigators (PIs) or collaborators contributing data to the ZIKV-IPD-MA members have provided declarations of potential or perceived conflict of interests in online supplemental appendix 3.

**Patient and public involvement** Patients and/or the public were not involved in the design, or conduct, or reporting, or dissemination plans of this research.

**Patient consent for publication** Not applicable.

**Ethics approval** The IPD-MA protocol was deemed exempt from ethical review by the WHO Ethics Review Committee and the Emory University Institutional Review Board because the IPD-MA uses de-identified, participant-level data from studies that have received ethical approval for the same objectives as the original studies. In keeping with recommendations for the re-use of de-identified participant-level data in an IPD-MA, we conducted a qualitative study to assess source communities' perceptions regarding the reuse of their data for an IPD-MA and preferences for the communication of ZIKV-related risk in the presence of uncertainty in both the diagnosis and the probability of ZIKV-related adverse events, given a positive diagnosis.

**Provenance and peer review** Not commissioned; externally peer-reviewed.

**Data availability statement** Key metadata are presented in the tables in the manuscript and corresponding appendices. Requests for access to the full metadata dataset for the 54 contributing studies should be made to the corresponding author. Not applicable.

**ORCID iDs**
Mabel Carabali http://orcid.org/0000-0002-9171-0483
Lauren Maxwell http://orcid.org/0000-0002-0777-2092
Brooke Levis http://orcid.org/0000-0002-4310-3689
Priya Shreedhar http://orcid.org/0000-0002-1920-2636

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
