## [Reviewer comments · BMJ Open]

ARTICLE DETAILS

TITLE (PROVISIONAL)	Heterogeneity of Zika Virus Exposure and Outcome Ascertainment Across Cohorts of Pregnant Women, their Infants, and their Children: A Metadata Survey
AUTHORS	Carabali, Mabel; Maxwell, Lauren; Levis, Brooke; Shreedhar, Priya

VERSION 1 – REVIEW

REVIEWER	R Huits Institute of Tropical Medicine, Antwerpen
REVIEW RETURNED	13-Jun-2022

GENERAL COMMENTS	In manuscript bmjopen-2022-064362, entitled 'Heterogeneity of Zika Virus Exposure and Outcome Ascertainment Across Cohorts of Pregnant Women, their Infants, and their Children: A Metadata Survey', the authors present document the presence of cross- and within study-level heterogeneity across 54 different ZIKV-IPD-MA participant studies from 20 countries or territories. Unsurprisingly, sources of heterogeneity across and within studies were identified 'that are not commonly observed in IPD-MAs of well characterized pathogens or in chronic diseases. The authors provide a well written and concise account of a real-life problem, i.e., study-level heterogeneity in the dynamics of research initiatives following the rapid spread across many countries of a pathogen like Zika virus. While the manuscript can inform ongoing and future collaborations, I am not convinced that it offers solutions to the issues of methodological diversity in multiple, simultaneously initiated studies of a rapidly emerging, previously poorly documented pathogen.
--

REVIEWER	Olivier Mukuku Institut Supérieur des Techniques Médicales de Lubumbashi, Research
REVIEW RETURNED	24-Jun-2022

GENERAL COMMENTS	Dear Editor, Thank you for this opportunity to review this manuscript. It is a very well-designed, presented, and structured work that deserves to be published in your journal. The authors did excellent work.
---

REVIEWER	Tara Kerin UCLA, Department of Pediatrics
REVIEW RETURNED	11-Oct-2022

GENERAL COMMENTS	1) "ZIKV IPD Consortium members formed expert panels that reviewed the exposure and outcome-related components " Can you be more clear on what constitutes an "expert" for this purpose? And
--

	how were the exposure and outcomes systematically reviewed? 2) "Investigators received two monthly reminders to complete the survey and were contacted individually to resolve study-level missingness and to address inconsistencies between survey responses and published study reports." How many investigators failed to return the survey (only 5?), and/or had significant amounts of unresolved missing data (same 5)? 3) " The difference between the number of study sites and unique datasets is explained by the inclusion of studies using data from a single surveillance system or multisite studies for which the metadata". This was explained very well, Thank you. 4) Under the "Quality of data and risk of bias Assessment" section, could you be more specific about the places and types of bias (were they more prevalent in certain regions), and also further explain potential (or really lack of potential) misclassification? 5) Overall, I thought this was a very thorough and complete meta analysis. I would like to see more text explaining the tables and figures to provide a more concise highlight of these additions. The supplement appendices were extremely detailed which is great, but it would be nice to also have more direct information regarding these in the text of the paper.
--	--

VERSION 1 – AUTHOR RESPONSE

Reviewer: 1

Dr. R Huits, Institute of Tropical Medicine, Antwerpen

Comments to the Author:

In manuscript bmjopen-2022-064362, entitled 'Heterogeneity of Zika Virus Exposure and Outcome Ascertainment Across Cohorts of Pregnant Women, their Infants, and their Children: A Metadata Survey', the authors present document the presence of cross- and within study-level heterogeneity across 54 different ZIKV-IPD-MA participant studies from 20 countries or territories.

Unsurprisingly, sources of heterogeneity across and within studies were identified 'that are not commonly observed in IPD-MAs of well characterized pathogens or in chronic diseases.

The authors provide a well written and concise account of a real-life problem, i.e., study-level heterogeneity in the dynamics of research initiatives following the rapid spread across many countries of a pathogen like Zika virus. While the manuscript can inform ongoing and future collaborations, I am not convinced that it offers solutions to the issues of methodological diversity in multiple, simultaneously initiated studies of a rapidly emerging, previously poorly documented pathogen.

Response: We would like to thank the reviewer for their thoughtful remarks. As indicated in the introduction, we have documented the sources of heterogeneity across included studies. These sources of heterogeneity (e.g., in screening algorithms, diagnostics, exposure and outcome definitions) can be expected to affect any meta-analysis of an emerging pathogen and their documentation is, in itself, a valuable contribution to the literature.

Given the descriptive nature of this analysis, offering specific solutions is beyond the scope of this manuscript. Nonetheless, as is also mentioned in the introduction, this analysis is part of a larger project, both in scope and magnitude, which will apply the key findings related to study-level heterogeneity from this formative work when combining participant-level data across Zika virus (ZIKV)-related cohorts of pregnant women and their infants and children.

The ZIKV Individual Participant Data (IPD) Consortium (ZIKV IPD-MA Consortium) is a large consortium of data-contributing groups and statistical methodologists that aims to extend the methods commonly applied in IPD-MA of well-understood exposures (e.g. HIV, heart disease) to account for the high level of heterogeneity and measurement error which affect any IPD-MA of an emerging

pathogen. Specifically, the ZIKV IPD Consortium-led IPD-MA brings together statistical methodologists who develop Bayesian approaches to modelling complex forms of measurement error, as when the measurement of the exposure is related to the probability of experiencing the outcome, with statistical methodologists who focus on leveraging IPD for improved diagnosis and prediction (1, 2). While we cannot provide solutions to these formidable challenges in the current manuscript, which focuses on describing the study-level sources of heterogeneity, we can point the reviewer to our prior work, where we describe these methodological challenges and further detail our work within the ZIKV IPD Consortium to develop the methods needed to address these challenges to inference in the context of IPD-MAs of emerging pathogens. That said, in response to the reviewer's critique, we have rephrased some sentences in the introduction and discussion sections as follows:

Introduction section: Page 6; last paragraph: "The in-depth evaluation of cross-study heterogeneity in exposure, outcome, and covariate ascertainment and definitions will be used to inform and contribute to the decisions about the analytic methods required to appropriately separate measurement- versus clinically derived sources of heterogeneity in the ZIKV IPD-MA. This comprehensive metadata review provides an outlook of different sources of bias and heterogeneity in the analysis of ZIKV-related studies and serves as a template for future work to harmonize and analyze IPD collected in the research response to emerging pathogens, where changes to best practice, definitions, and study and laboratory protocols proscribe the use of traditional metadata extraction."

Discussion section: Page 15/16; last paragraph "

The ZIKV-IPD Consortium recognized that existing methods for addressing the correlated sources of heterogeneity in the research response to ZIKV (e.g., dependency between timing of symptoms, test accuracy, and differential ascertainment for both exposure and outcome, etc.) are inadequate. In response to this gap, the Consortium launched a statistical WG to develop novel approaches to address heterogeneity and measurement error in the context of IPD-MAs of emerging pathogens. The statistical WG will use the findings from this metadata survey to guide the development of an analytic plan that accounts for the different sources of correlated measurement error. The statistical analysis plan will extend existing methods for measurement error and clinical risk prediction to address the specific challenges of an IPD-MA of an emerging pathogen. Methodological approaches for accounting for correlated sources of measurement error and the high levels of heterogeneity in the research response to emerging pathogens will be context- and pathogen-specific. Although specific recommendations or solutions to account for the diverse sources of heterogeneity are desirable, an in-depth proposal for the methods to address these statistical challenges is outside of the scope of this descriptive analysis."

References:

1. Zika Virus Individual Participant Data Consortium. The Zika Virus Individual Participant Data Consortium: A Global Initiative to Estimate the Effects of Exposure to Zika Virus during Pregnancy on Adverse Fetal, Infant, and Child Health Outcomes. *Trop Med Infect Dis.* 2020 Sep 30;5(4):152. doi: 10.3390/tropicalmed5040152.
2. Wilder-Smith A, Wei Y, ... Zika Virus Individual Participant Data Consortium, et al. Understanding the relation between Zika virus infection during pregnancy and adverse fetal, infant and child outcomes: a protocol for a systematic review and individual participant data meta-analysis of longitudinal studies of pregnant women and their infants and children. *BMJ Open* 2019;9 : e026092. doi: 10.1136/bmjopen-2018-026092

Reviewer: 2

Dr. Olivier Mukuku, Institut Supérieur des Techniques Médicales de Lubumbashi, Université de Lubumbashi Faculté de Médecine

Comments to the Author:

Dear Editor,

Thank you for this opportunity to review this manuscript. It is a very well-designed, presented, and structured work that deserves to be published in your journal. The authors did excellent work.

Response: Thank you for your remarks and kind consideration.

Reviewer: 3

Dr. Tara Kerin, UCLA

Comments to the Author:

1) "ZIKV IPD Consortium members formed expert panels that reviewed the exposure and outcome-related components" Can you be more clear on what constitutes an "expert" for this purpose? And how were the exposure and outcomes systematically reviewed?

Response: Thank you for this excellent suggestion. We have added further details to the section describing the panel and the way in which the exposures and outcomes were reviewed as follows:

Methods: Metadata survey development and dissemination section; page 7/8: "ZIKV IPD Consortium members formed expert panels that were designated as subject matter working groups (WGs) for outcomes, exposures, social sciences, metadata and harmonization. Each WG included clinicians with expertise in relevant areas (e.g., neonatology, neurology, psychology, virology), epidemiologists, sociologists, and members of the metadata and harmonization team (i.e., consortium members who will work on the harmonization of the data as it becomes available). Using the ZIKV IPD-MA protocol as a guide, the WGs met and reviewed the established exposure and outcome-related components of the metadata survey to ensure its comprehensiveness, this is, that the draft survey included all potential exposures, outcomes, and likely confounders, and the methods for their ascertainment or measurement. The WGs also assessed the utility of survey in terms of public health and epidemiological relevance to potential end users such as other researchers, policy makers, and health care practitioners."

2) "Investigators received two monthly reminders to complete the survey and were contacted individually to resolve study-level missingness and to address inconsistencies between survey responses and published study reports." How many investigators failed to return the survey (only 5?), and/or had significant amounts of unresolved missing data (same 5?)?

Response: Yes, five (n=5) studies did not initiate or complete the metadata survey. From those, the provided non-mutually exclusive reasons for non-participation were inadequate time or resources to support participation (n=4), lack of consensus on participation from their team (n=1), and need for special permissions from MoH (n=1). These non-participant studies did not provide data at the time of the analysis and therefore information of unresolved queries or missing data could not be evaluated. We added the details in the text as follows:

Results section; page 9: "Remaining studies (n=5) did not initiate or complete the metadata survey after nine months of regular follow-up. The four teams that reported a reason for non-participation, reported multiple considerations including inadequate time or resources to support participation (n=4), lack of consensus on participation from their team (n=1) and need for special permissions from MoH (n=1).."

3) " The difference between the number of study sites and unique datasets is explained by the inclusion of studies using data from a single surveillance system or multisite studies for which the metadata". This was explained very well, Thank you.

Response: Thank you, we tried to clarify this as much as possible.

4) Under the "Quality of data and risk of bias Assessment" section, could you be more specific about the places and types of bias (were they more prevalent in certain regions), and also further explain potential (or really lack of potential) misclassification?

Response: Thank you for this remark. We have further clarified and added the details concerning the risk of bias as follows:

Results: Quality of data and risk of bias assessment section; page 13: "The potential sources of bias identified across the participant studies relate to selection bias due to differential selection, as by the presence of symptoms or observable congenital malformations, into the study and follow-up, and measurement error due to exposure and outcome ascertainment which were differential by the presence of symptoms and by ZIKV assay results. Hence, the potential risk of both selection bias and measurement error was likely highest at the beginning of the epidemic and surveillance-based studies were likely affected by under and overreporting at different stages of the epidemic. The metadata survey demonstrated that study protocols changed importantly over the course of study implementation to account for advances in the understanding of ZIKV presentation (e.g., the high percentage of asymptomatic cases), diagnostics (e.g., cross-reactivity with DENV), which increased the likelihood for potential misclassification, and for the ascertainment of long term outcomes (e.g., development of adverse developmental outcomes that were not observed at birth), and to accommodate funding availability, especially in the Americas region. The misclassification of ZIKV infection is foregrounded in our presentation of the diversity of methods, resources, definitions, and classifications applied in studies' ascertainment of maternal, fetal, and infant ZIKV infection across all study settings and designs (Table 1, Figure 4 and 5). Genetic testing, fetal ultrasounds, and MRIs used for outcome ascertainment is presented in Appendix 1, Table S4, and the type of maternal, fetal and infant ZIKV exposure assessment among ZIKV-IPD-MA study participants of the metadata survey are presented in Appendix 1, Tables S5a-S5c.

5) Overall, I thought this was a very thorough and complete meta analysis. I would like to see more text explaining the tables and figures to provide a more concise highlight of these additions. The supplement appendices were extremely detailed which is great, but it would be nice to also have more direct information regarding these in the text of the paper.

Response: Thank you for this remark. We have further highlighted key relevant results of our metadata analysis, and have clarified the location and utility of some of the appendices as follows:

Page 7: "Details of the systematic search to identify eligible studies, including the search strategy, PICO criteria, eligibility, and the screening and selection steps are presented in Appendix 1, Figure S1. Eligible studies were invited via e-mail to share data and to actively participate in the development and completion of the IPD-MA."

Page 7: "we developed a comprehensive metadata survey of completed and ongoing ZIKV-related studies, available in English, Spanish and Portuguese (Appendix 2)."

Page 9/10: "Details including study site, authors, enrollment characteristics, study design, recruitment, follow-up, sample size and type of funding are presented in Appendix 1, Table S1."

Page 13: "Genetic testing, fetal ultrasounds, and MRIs used for outcome ascertainment is presented in Appendix 1, Table S4, and the type of maternal, fetal and infant ZIKV exposure assessment among ZIKV-IPD-MA study participants of the metadata survey are presented in Appendix 1, Tables S5a-S5c."

VERSION 2 – REVIEW

REVIEWER	Tara Kerin UCLA, Department of Pediatrics
REVIEW RETURNED	28-Oct-2022
GENERAL COMMENTS	Thank you for addressing clarification points. This is a well-done and well-organized paper.